# The Role of Dietary Fibers in the Management of IBD Symptoms

**DOI:** 10.3390/nu14224775

**Published:** 2022-11-11

**Authors:** Claudia Di Rosa, Annamaria Altomare, Elena Imperia, Chiara Spiezia, Yeganeh Manon Khazrai, Michele Pier Luca Guarino

**Affiliations:** 1Research Unit of Food Science and Human Nutrition, Department of Science and Technology for Humans and the Environment, Università Campus Bio-Medico di Roma, Via Alvaro del Portillo, 21, 00128 Roma, Italy; 2Research Unit of Gastroenterology, Department of Medicine, Università Campus Bio-Medico di Roma, Via Alvaro del Portillo, 21, 00128 Roma, Italy; 3Operative Research Unit of Gastroenterology, Fondazione Policlinico Universitario Campus Bio-Medico, Via Alvaro del Portillo, 200, 00128 Roma, Italy; 4Operative Research Unit of Nutrition and Prevention, Fondazione Policlinico Universitario Campus Bio-Medico, Via Alvaro del Portillo, 200, 00128 Roma, Italy

**Keywords:** inflammatory bowel disease, dietary fibers, nutrition

## Abstract

Inflammatory bowel diseases (IBDs) are chronic, progressive, immune-mediated diseases of the intestinal tract. The main subtypes of IBDs are Chron’s disease (CD) and ulcerative colitis (UC). The etiology is still unclear, but there are genetic, environmental and host-related factors that contribute to the development of these diseases. Recent literature has shown that dietary therapy is the cornerstone of IBD treatment in terms of management of symptoms, relapse and care of the pathology. IBD patients show that microbiota dysbiosis and diet, especially dietary fiber, can modulate its composition. These patients are more at risk of energy protein malnutrition than the general population and are deficient in micronutrients. So far, no dietary component is considered responsible for IBD and there is not a specific therapeutic diet for it. The aim of this review is to evaluate the role of dietary fibers in CD and UC and help health professionals in the nutritional management of these pathologies. Further studies are necessary to determine the appropriate amount and type of fiber to suggest in the case of IBD to ameliorate psychosocial conditions and patients’ quality of life.

## 1. Introduction

Inflammatory bowel diseases (IBDs) are chronic, progressive, immune-mediated diseases of the intestinal tract [1] characterized by a progressing or relapsing and remitting disease course [2]. They are associated with significant morbidity, disability, and risk of complications [1] such as abdominal abscesses, fistulae, strictures and subsequent bowel obstruction, and an increased risk of gastrointestinal (GI) cancer [2]. These diseases have a significant impact on patients’ quality of life and on daily living activities and also increase health care costs [2]. IBD prevalence has increased from 79.5 to 84.3 per 100.000 people from 1990 to 2007 [3] and its prevalence rate varies substantially across countries—in fact, it is estimated that 1.3 million of people are affected in Europe [3].

To date, the etiology of these diseases is still not clear [4]; nevertheless, it is known that there are genetic [5,6,7,8,9,10,11,12,13,14,15,16,17], host-related [18,19,20,21,22] and environmental [16,19,23] factors that contribute to the development of gut inflammation and IBD (Table 1).

The main subtypes of IBDs are Crohn’s disease (CD) and ulcerative colitis (UC) [4]. CD inflammation affects the whole intestine but the most common sites are in the small and large intestine (especially in the ileum) and perianal region and are classified as “L1: ileal-type”, “L2: colonic-type”, “L3: ileocolonic-type” and “L4: isolated upper disease” [20]. UC, on the other hand, is classified as: E1 ulcerative proctitis, when the site of the disease is limited to the rectum; E2 distal UC, when it is limited to a portion of the colorectal distal to the splenic flexure; and E3 extensive UC or pancolitis, when it extends from proximal to the splenic flexure [24]. The severity of IBD is classified as “remission”, “mild”, “moderate”, or “severe” based on clinical symptoms, signs, and blood tests [24]. Signs and symptoms of CD and UC are presented in Table 2.

For IBD, there are both pharmacological and non-pharmacological options to manage symptoms. Current drug treatments for IBD are aminosalicylates (5-ASA) that can be used in combination with steroids to induce and maintain remission [28,29]. If CD is mild to moderate, mesalamine is the first-line therapy while sulfasalazine is most effective at maintaining remission in UC [28,30]. The mainstay treatments of IBDs are hydrocortisone and prednisolone with glucocorticoid properties [28]. The preferred steroid is prednisolone, administered orally [28] (0.75–1 mg/kg of body weight) [5] and can be used in the short term to alleviate symptoms of moderate and severe CD [28]. Moreover, with a low dose of steroids, azathioprine may be introduced [28]; in fact, immune suppressants have been adapted for the treatment of IBD. Thiopurines (azathioprine, 6-mercaptopurine and methotrexate) are beneficial in 50 to 70% of patients [28] but are generally reserved for patients who are steroid resistant or steroid dependent [28,31]. Anti-TNF monotherapy is effective in maintaining remission [28], especially in patients with moderate–severe UC who have inadequate response or intolerance to conventional therapy [28,29].

A treatment which can prevent inflammation remains a cornerstone for IBD management and the diet also plays a pivotal role in the prevention of inflammatory bowel disease risk development [32,33]. The spread of the Western diet, rich in fats, protein, simple sugars and low in fruit and vegetables, represents a possible cause for the increase in IBD [34]. Indeed, diet affects intestinal inflammation through mechanisms such as the ability to present antigens and the alteration of the balance of prostaglandins (involved in the inflammatory pattern) and microbiota [35,36]. In particular, numerous studies have underlined the increased incidence of IBD due to excessive consumption of sugars or sugar-sweetened beverages [19,37], proteins (especially from red meat) [38], animal fats and linoleic acid. In fact, red meat consumption has been suggested to have a pro-inflammatory effect. This may be due to the cooking method and the concurrent presence of saturated fats that establish deleterious effects [19]. A high consumption of total fatty acids, polyunsaturated fatty acids (PUFAs), especially omega 6 fatty acids, increases the risk of developing both UC and CD [23]. The recent literature also shows that nutrition could influence the development of IBD [39] but actually few dietary recommendations exist. The *American Journal of Gastroenterology* published the American College of Gastroenterology (ACG) Practice Guidelines for CD and UC, which suggest eating frequent small meals or snacks every 3–4 h, drinking enough fluids to avoid dehydration, eating food with added probiotics and prebiotics and using multivitamins [40]. During remission, patients can include whole grains and a variety of fruits and vegetables in their daily eating plan; however, when there are symptoms, it is recommended to consume low-fiber foods [40]. Foods recommended for IBD are: low-fat and lactose-free milk and dairy products, lean and white meat, fish or eggs, grains with less than 2 g of fiber per serving, low-fiber vegetables and fruits (i.e., lettuce, strained vegetable juice, fruit juice without pulp, ripe banana or melons), less than eight teaspoons/day of oils and to drink water or beverages without caffeine [40]. The effect of alcohol in IBD is still controversial: some studies document positive effects of alcohol consumption in the development of UC [41,42] but this effect seems to be nullified if alcohol consumption is associated with cigarette smoking [43].

The aim of the present review is to evaluate the current role of dietary fibers in both CD and UC to help health professionals in the nutritional management of these pathologies in order to lead patients to choose or to avoid fiber-containing foods depending on the stage of their disease.

## 2. Fibers Classification and Functions

There are different definitions of dietary fiber collected over the years and, currently, there is not only a single accepted definition. The term “fiber” was coined for the first time in 1953 [44], but it has evolved over time. According to the American Association of Cereal Chemists (AACC), dietary fiber is “the edible parts of plants or analogous carbohydrates that are resistant to digestion and absorption in the human small intestine, with complete or partial fermentation in the large intestine”. Thereafter, in 2007, FAO/WHO experts defined the term fiber as “non-digestible carbohydrates contained in cereals, seeds, vegetables and fruit” [45]. To date, this last definition is accepted internationally. However, in 2009, the Codex Alimentarius defined dietary fiber as “carbohydrate polymers with ten or more monomer units that are not hydrolyzed by endogenous enzymes in the human small intestine” [46]. In the end, a physiological description of this element consists of: “dietary components that are not enzymatically broken down into absorbable subunits in the stomach and small intestine” [47].

Dietary fibers can be classified, based on their water solubility, into insoluble (IDF) and soluble dietary fibers (SDFs) [48]. IDFs are not soluble in water and present reduced fermentability [49]. This type of fiber is mostly present in plants and it is a structural component of cell walls. It includes cellulose, water-insoluble hemicellulose and lignin [50]. Wholemeal flour, wheat bran, brown rice, nuts, beans, vegetables and their peels (such as cabbage, celery, cauliflower, and Brussels sprouts) and fruit contain large doses of IDF [49].

IDFs have a laxative action and increase the sense of satiety; in fact, their consumption helps to reduce caloric intake and to control body weight [51,52].

On the other hand, the main characteristics of SDFs are: solubility in water, ability to form viscous solutions and fermentability [49]. They consist of a variety of non-cellulosic polysaccharides and oligosaccharides such as pectins, β-glucans and water-soluble gums [53]. This type of dietary fiber is abundant in whole grains (e.g., oats, barley, and wheat), legumes (e.g., lentils, split peas, guar seeds, pinto beans, black beans, red beans, chickpeas and lima beans), some fruits and vegetables (apples, oranges and carrots) and seeds (e.g., linseed and psyllium seeds) [49]. SDFs contribute to the reduction in blood lipid levels, blood pressure profile, inflammation, risk of cardiovascular disease (CVD) and intestinal transit time, while on the other side, they determine an improvement in blood sugar levels, weight control, immune function and short-chain fatty acids (SCFAs) levels [54,55,56,57,58].

The European Food Safety Authority (EFSA) recommends that healthy adults should consume a minimum of 25 g of fiber per day to ensure adequate intestinal functions [59]. The American Heart Association Eating Plan suggests a varied diet to guarantee assumption of fibers from a variety of sources. Total dietary fiber intake should be from 25 to 30 g/day from food, without any supplements [60]. As general recommendations, the position of the Academy of Nutrition and Dietetics [61] suggests consuming an adequate amount of fiber from a variety of plant food sources. Based on studies in the literature evaluating protection against coronary heart disease, an adequate fiber intake is 14 g of total fiber per 1.000 kcal, or 25 g for women and 38 g for men [61]. This amount reduces the risk of several chronic diseases, such as CVD, type 2 diabetes, and certain types of cancer, limiting inflammation and modulating the immune response [61,62].

## 3. The Effect of Dietary Fibers in IBD

In the literature, since 1978, there has been considerable interest in dietary fibers as a therapeutic option in IBD but its role is unclear, with conflicting findings [63].

In the case of IBD, it is suggested that a diet rich in fibers, vitamin D and adequate consumption of citrus fruits has a protective role [39]. In fact, high consumption of fiber and fruit is associated with a 73–80% decreased risk of CD, while a high intake of vegetables reduces the risk of UC [23].

It is important to underline that patients with stricturing Crohn’s disease should be careful to their intake of dietary fiber and fibrous foods for symptomatic management of strictures and may need supplementation with enteral or parenteral nutritional requirements [64].

Although Anantakrishnan et al. observed that the consumption of at least 24.3 g of fiber per day (particularly fruit-derived fiber) reduces the risk of developing Crohn’s disease by 40% but not UC [65], in the literature, there are some conflicting evidence on the use of high-fiber foods in IBD. To date, it is clear that they are not recommended during flare-ups or during active disease states, fistulas or strictures [40]. In fact, in CD, a low-fiber diet should be used for a short period, and it is indicated only in certain conditions, such as acute relapses (with diarrhea and cramps), intestinal stenosis, bacterial proliferation of the small intestine and after some type of surgery [40]. In 2020, Day et al., in a systematic review, analyzed different study designs to determine the correct amount of dietary fiber in individuals with IBD [66]. They compared the adequacy of fiber intakes with that of a control group or compared to national dietary guidelines, and examined factors associated with fiber consumption [67]. They concluded that individuals with IBD are used to consuming less fiber than healthy populations and that fiber intakes are inadequate compared to national fiber guidelines [66].

It is also necessary to mention the role of fermentable carbohydrates, or FODMAPs (oligosaccharides, disaccharides, monosaccharides and fermentable polyols), in IBD. FODMAPs have an osmotic and fermentative action because they recall water and gas in the intestinal lumen being partially or completely indigestible [68]. A low-FODMAP diet is usually suggested in the case of Irritable Bowel Syndrome (IBS) that is characterized by symptoms such as abdominal pain, meteorism, bloating, diarrhea or constipation [69,70]. Approximately one-third of IBD patients complain of IBS-like intestinal symptoms in the absence of actual gastrointestinal inflammation, thus usually an overlap of symptoms has been recognized [71]. There are several studies that evaluate the association between IBD and consumption of FODMAPs. In a randomized controlled trial (RCT), improvements in gastrointestinal symptoms were observed in patients in remission or with mild–moderate disease and coexisting IBS-like symptoms after following a low-FODMAP diet for approximately 6 weeks [72]. Results showed a significant reduction in symptoms in the Low-FODMAP diet group compared to the No Diet group (*p* = 0.02) [72]. A further study evaluated the effect of FODMAP consumption in quiescent IBD patients allocated to a series of 3-day fermentable carbohydrate challenges in a random order (fructan: 12 g/die; galacto-oligosaccharides: GOS, 6 g/die; sorbitol: 6 g/die; and glucose placebo: 12 g/die), each separated by a washout period [73]. At the end of the study, the authors observed that fewer patients reported adequate relief of functional gastrointestinal symptoms (62.1%) compared to glucose (89.7%) (*p* = 0.033), and that fructans, but not GOS or sorbitol, exacerbated gastrointestinal functional symptoms [74]. However, it is necessary to carry out further studies to evaluate the effect of different types and doses of FODMAP in patients with IBD [73].

## 4. Side Effects of Dietary Fibers with a Focus on IBD

To date, the daily amount of dietary fiber is suggested by the EFSA, The American Heart Association Eating Plan and the Academy of Nutrition and Dietetics [59,60,61] but there is no tolerable upper limit for total fiber intake; thus, eating more fibers can cause uncomfortable side effects [75], such as constipation, intestinal blockage, bloating or diarrhea [76,77,78]. Inadequate hydration and reduced physical activity combined with high fiber consumption can induce constipation; fiber is not fully digested or excreted from the body, generating intestinal blockages. In this case, intestinal bacteria ferment these undigested fibers [79], producing gas [80] and leading to symptoms such as bloating and flatulence. One strategy to minimize intestinal gas production is to consume fibers gradually and not suddenly. On the other hand, excessive consumption of fibers, especially insoluble ones, can also lead to diarrhea [75,81].

Moreover, high fiber consumption can reduce the rate of gastrointestinal micronutrient absorption, especially calcium and iron [82]. This is due to the chemical and physical characteristics of fiber such as fermentation, bulking capacity, binding capacity, viscosity and gel formation, water retention capacity and solubility [83] and also to the presence of some compounds such as phytates, oxalates, and tannins [84]. This is a big issue for IBD patients, who are already malnourished for gastro-intestinal malabsorption, and are further destabilized.

IBD patients report intolerance to some types of fibers because they lack fermentative microbe activities compared to individuals with normal microbial fermentative activity [84]. The absence of these microbes makes the fibers indigestible and therefore intact; this fibers interact with host cell receptors and promote intestinal inflammation [84].

The study by Armstrong et al. investigated the role of unfermented β-fibers in fueling inflammation in IBD patients [85]. β-Fructans (inulin and oligofructose/FOS) are β-(2 → 1)-linked fructose oligo- and polysaccharides. They are abundant in plant sources such as chicory root, agave and artichokes, while less abundant in banana, wheat, onion and garlic [86].

In some IBD patients, β-fructan fibers have shown potential negative impacts. Indeed, dietary β-fructans induce inflammation through activation of TLR2 and NLRP332 pathways [87] and promote the production of reactive oxygen species (ROS) interacting with carbohydrate receptors (GLP-1R) [88,89].

Although inulin could present positive effects on inflammation, there are several studies that have shown that it exacerbates the severity of colitis in an IL10-/- and DSS model of colitis [90] and facilitates the progression of hepatocellular carcinoma in mice [91].

Moreover, it has been seen that the consumption of unfermented FOS could also induce the production of pro-inflammatory cytokines in peripheral blood mononuclear cells (PBMCs) and in THP-1 macrophages, and this was seen in biopsies from IBD patients [92,93].

## 5. Gut Microbiota in IBD

In IBD patients, there is dysbiosis of the gut microbiota, which is reduced in richness and diversity compared to microbiota of healthy individuals, enrichment of *Enterobacteriaceae* and a reduced *Firmicutes* and *Bacteroides* ratio [94,95,96,97]. There is a reduction in *Lachnospiraceae* and *Bacteroidetes* and an increase in *Veillonellaceae*, *Fusobacteriaceae*, Proteobacteria (i.e., *Enterobacteriaceae* or *Klebsiella pneumoniae*) and Actinobacteria [98,99]. Federici et al. showed that *K. pneumoniae* (Kp) strains are associated with IBD severity across, thus Kp strains were isolated and animal IBD models were colonized by Kp to induce gut inflammation. In this way, they were able to develop a phage combination therapy that specifically suppressed the pathogenic Kp2 clade and intestinal inflammation in IBD models [100].

Moreover, Actinobacteria are potentially pathogenic species that, in specific conditions, could produce a large amount of toxins provoking the activation of intestinal inflammation responsible for the onset of IBD symptoms. In CD, *Firmicutes*, in particular *Clostridium leptum* and *Faecalibacterium prausnitzii* [98], are reduced compared to healthy people. *F. prausnitzii* has an anti-inflammatory action; in fact, it is one of the butyrate-producing bacteria that contributes to maintaining the integrity of mucosa and reduces the adhesion and colonization of pathogens in the intestinal tract [98]. On the contrary, *Prevotella* is increased [98]. *Prevotella* is able to degrade mucin glycoproteins of the gut mucosal layer, so intestinal barrier function is altered, its permeability is increased and there is a significant translocation of pathogens [99]. In UC, however, the amount of *Akkermansia* (*A.*) *muciniphila* is reduced [101]. *A. muciniphila* is a SCFA-producing bacteria and its function also involves the degradation of the mucus layer, converting the mucin in host beneficial products [101,102].

Diet modulates and supports the growth, the diversity and the richness of the gut microbiota. Type, quality and origin of food influence the microbial community by altering host–microbe interaction [103]. In fact, a low intake of dietary fibers and increased amounts of fat and sugar, typical of a Western diet, may contribute to depletion of specific bacterial taxa [103]. Dietary fibers play a pivotal role in modulating the gut microbiota as it regulates macronutrient metabolism and host physiologic conditions [103].

## 6. Mechanism of Action of Dietary Fibers in IBD

Although current dietary guidelines are not so clear on the amount and the type of fiber that should be consumed in IBD [67], in the literature, several studies evaluated the effects of fermentable and non-fermentable fibers in IBD patients.

It is clear that some fermentable fibers such as resistant starch (that is not digested in the small intestine) and inulin are metabolized by intestinal bacteria to SCFAs [74], acetate, butyrate and propionate, and they have immunomodulatory properties, promote the regeneration of the intestinal epithelium, lower the pH of the colon and inhibit the growth of pathogens [104].

The IBD-altered microbiota composition results in a lower production of anti-inflammatory and immunoregulatory metabolites, in particular butyrate, a lack of which may contribute to increase intestinal inflammation [19]. Butyrate plays a central role in the development of IBD because it represents the main energy substrate for colonocytes [105] and the alteration of its metabolism is linked with mucosal damage and inflammation [106]. In fact, the integration of some types of fibers (especially fermentable fibers) produces SCFAs capable of maintaining remission and reducing mucosal lesions [104]. In addiction, SFCAs, particularly acetate and butyrate, balance mucus production and secretion. Mucus production at the level of the epithelium is a form of host protection to prevent microbial invasion and susceptibility to infection [107]. A diet low in fiber produces less SFCAs and results in an increase in harmful metabolites that increases susceptibility of infections by deterioration of the mucus layer [108] and contribute to the development of chronic disease and colorectal cancer (CRC) [107].

In UC patients in remission, different types of fibers have been tested. In fact, Davies and Rhodes in 1978 [109] evaluated the effect of oat bran that contains insoluble and non-fermentable fiber. It favored the stool increase, but did not determine a huge production of butyrate. Different results have been observed by Hallert et al. He described the supplementation of 60 g per day of oat bran (20 g of dietary fiber) to the daily diet in 22 patients with UC remission, mainly as bread slices, for three months. In this pilot study, the authors noticed an increase in fecal butyrate production and a concomitant reduction in gastrointestinal symptoms. He stated that a diet rich in oat bran is safe in patients with quiescent UC [110]. In a previous study, Hallert and his group [111] evaluated the gastrointestinal effects of *Plantago ovata* peel, composed mainly of soluble and fermentable fibers. After 4 months of intervention, 69% of the participants showed relief of symptoms. This effect is attributable to the type of fiber used which increased the production of luminal SCFAs in the ascending colon. Subsequently, Fernandez-Bañares et al. in 1999 [112] administered *Plantago ovata* seeds (that contain both insoluble and soluble fiber) to 105 patients in UC remission (102 in final analysis). They were randomized into three groups and received: *Plantago ovata* seeds (10 g twice daily), mesalamine (5-aminosalicylate derivative used as therapy to maintain remission) and plantago ovata seeds + mesalamine with the aim to maintain the remission until 12 months. At the end of this pilot study, the *Plantago ovata* seed group showed an increase in butyrate production (*p* = 0.018) but none of these three treatments reached the goal. In fact, after approximately 9–10 months, there was a disease relapse in all groups. This result does not mean that all treatment are equivalent but that, unfortunately, none of them succeeded in maintaining the remission for 12 months [112].The seeds of *Plantago ovata* are therefore able to produce SCFAs, especially butyrate, in the distal parts of the colon, and not only in the proximal colon, as in the case of the peel of *Plantago ovata*, mainly composed of fermentable soluble fiber [92]. Moreover, also Germinated Barley Foodstuff (GBF), [113,114,115,116,117] a fiber derived from the aleurone layer and the scutello fractions of the spent beer kernels, was tested as nutritional treatment. GBF fiber is composed by slightly lignified hemicellulose and it is used by *Bifidobacterium* and *Lactobacillus* to produce SFCAs, especially butyrate, in the lumen of the colon [113]. Numerous studies in the literature have evaluated positive effects of GBF and found no side effects in UC patients. In fact, supplementation with GBF in UC patients determined an improvement in clinical activity and the endoscopic index scores and also a reduction in the dose of glucocorticosteroids taken and in the frequency of clinical signs (number of episodes of diarrhea, degree of visible blood in stools, degree of abdominal pain or cramps, nausea, vomiting and anorexia) [113,117,118]. Additionally, inulin plays a key role in UC. Wetters et al. evaluated the effect of inulin in 20 patients with chronic pouchitis after colectomy for UC in a crossover study. After administration of 24 g inulin/day, there was increased intestinal butyrate production, a lowered pH, and a decreased number of *Bacteroides fragilis*. Furthermore, at the endoscopic and histological levels, a reduction in inflammation of the ileal mucosa was observed [119]. Effects of dietary fibers in studies in UC are reported in Table 3.

For CD, the combination of inulin and oligofructose has shown some good results. In particular, Lindsay et al. administered 15 g/day of FOS as a supplement (to be dissolved in water or food) and it contained a mixture of oligofructose and inulin (ratio 70:30%) in 10 patients with active ileocolonic Crohn’s disease for 3 weeks. This study showed a significant increase in mucosal *Bifidobacteria*. There was also an increase in colonic dendritic cells expressing IL-10, the Toll-like receptor TLR-2 and TLR-4. The increase in these factors indicates that this type of prebiotic stimulates the mucosa innate immune response [120]. Detailed analyses of CD patients’ microbiota showed that bacteria such as *Ruminococcus gnavus* and *Bifidobacterium longum* play a central role in the development of dysbiosis. In a further randomized placebo-controlled trial on 67 patients with inactive and mild to moderately active CD the effect of oligofructose-enriched inulin (OF-IN) or placebo 10 g twice daily for 4 weeks was evaluated. The results showed both a decrease in *Ruminococcus gnavus* and an increase in the number of *Bifidobacterium longum* (*p* = 0.02). Furthermore, in the subgroup of patients with active CD, there was also a positive correlation between the increase in the number of *Bifidobacterium longum* and the improvement in disease activity [121] (Table 4).

## 7. Conclusions

Diet plays a crucial role in the treatment of IBD [122]; however, no dietary component is considered responsible for the disease [64]. Thus, patients with IBD should be advised to eat a varied diet that meets their energetic and nutrients requirements, including dietary fibers [64]. In the literature, it is clear that IBD subjects tend to consume less fiber than healthy controls [66]. Studies have shown that fiber supplementation alone is unlikely to restore IBD patients’ microbiota to a healthy state [123].

IBD patients are more at risk of protein-energy malnutrition than the general population. They have difficulty in gaining weight and, especially those affected by CD, could have deficiencies in micronutrients such as iron, vitamin B12 and vitamin D [124,125,126,127].

All of these nutritional problems also have a serious psychosocial repercussion and worsen patients’ quality of life [128]. In fact, a majority of individuals with IBD believe that specific foods trigger their disease flares, although this belief is not supported by any study [129].

Our review aimed to evaluate the effects of dietary fibers in the two different types of IBD, Crohn’s disease and ulcerative colitis. Actually, there is no consensus on the type and the amount of dietary fibers to suggest in these two cases even when taking into consideration the phase of the disease. Further studies are necessary to determine the appropriate amount and type of fiber to suggest in the case of IBD.

## Figures and Tables

**Table 1 nutrients-14-04775-t001:** Etiopathogenetic factors and its effects on IBD patients.

Etiopathogenetic Factors		Effects
Genetic factors	NOD2 gene mutation [5,6,7,8,9,10,11]	Alteration of intestinal immune homeostasis and components which maintain the mucus layer
	ATG16L1 gene mutation [5,6,12,13,14,15]	Paneth cell function in autophagy mechanisms is compromised, so protection against infection removing many intracellular microbes is reduced
	Locus IBD5 alteration [16,17]	Wrong codification of a group of cationic organic transporters, OCTN1 and OCTN2. Reduction in cells and tissues from oxidative and/or inflammatory damage
	Locus IBD3 alteration [16]	Wrong codification of Major Histocompatibility Complex (MHC)
Host-related factors	Microbiota alteration [18,19]	Lower production of anti-inflammatory and immunoregulatory metabolites, in particular butyrate—a lack of which may contribute to increased intestinal inflammation
	Immune response [18,20,21,22]	Hyperactivity of T cells with excessive production of cytokines, among which IL-12, il-23 and IFN-γ promote a TH1 and TH17 lymphocytic phenotype. The inhibition of the effector cytokines, such as TNF-α
Environmental factors	Diet [19,23]	Red meat consumption has a pro-inflammatory effect. A high consumption of total fatty acids, polyunsaturated fatty acids (PUFAs), especially omega 6 fatty acids, increases the risk of developing both UC and CD
	Cigarette smoking [16]	Formation of fistulas and intestinal strictures increases the frequency of exacerbations and favors post-surgical relapses in CD. On the contrary, in UC, it seems to have a protective action and it is associated with less frequent flare-ups of the disease

**Table 2 nutrients-14-04775-t002:** Symptoms and signs of CD vs. UC [25,26,27].

Symptoms and Signs	CD		UC	
	Presence/Absence	Frequency	Presence/Absence	Frequency
Abdominal pain	✔	+	✔	+
Diarrhea	✔	+	✔	+
Hematochezia	✔/✗	+/−	✔/✗	+
Abdominal mass	✔	+	✔	+/−
Malnutrition	✔	+	✔/✗	+/−
Abdominal distension	✔/✗	+/−	✔/✗	+/−
Sub occlusive symptoms	✔	+	✗	-
Perianal disease	✔	+/−	✗	-
Fistulas	✔	+/−	✗	-
Anemia	✔	+	✔/✗	+/−
Iron deficiency	✔	+	✔/✗	+/−
Low vitamin D	✔	+	✔/✗	+/−
Elevated inflammatory markers	✔	+	✔/✗	+/−

✔ the symptoms/sign is present; ✗ the symptoms/sign is absent; ✔/✗ is present occasionally or during the acute phase of disease; + frequent; +/− variable; - absent.

**Table 3 nutrients-14-04775-t003:** Effects of dietary fibers on UC subjects.

	Number of Patients	Duration	Type and Amount of Fiber	Results
Davies and Rhodes, 1978 [109]	39 subjects in UC remission	6 months	25 g/day Oat bran	Increased stool but no effects on butyrate production
Hallert et al., 2003 [110]	22 subjects in UC remission	3 months	60 g/day Oat bran	Increase in butyrate production and a decrease in gastrointestinal symptoms
Hallert et al., 1991 [111]	23 subjects in UC remission	4 months	Plantago ovata peel	69% of patients showed relief of symptoms for increased SCFAs production
Fernandez-Bañares et al. in 1999 [112]	105 subjects in UC remission	12 months	Arm 1: 10 g twice a day of plantago ovata seeds; Arm 2: mesalamine; Arm 3: plantago ovata seeds + mesalamine	The three arms showed the same results on symptoms
Mitsuyama et al., 1998 [113]	10 subjects with active UC	1 month	30 g/day GBF	Patients showed improvement in their clinical activity index scores, with a significant decrease in the score
Wetters et al., [119]	20 subjects	3 weeks	24 g inulin/day	Compared with placebo, inulin increased butyrate concentrations, lowered pH, decreased numbers of Bacteroides fragilis, and diminished concentrations of secondary bile acids in feces

**Table 4 nutrients-14-04775-t004:** Effects of dietary fibers on CD subjects.

	Number of Patients	Duration	Type and Amount of Fiber	Results
Lindsay et al. [120].	10 subjects with ileocolonic CD	3 weeks	15 g/day FOS (70:30% oligofructose:inulin)	Increase in mucosal Bifidobacteria, in IL-10, TLR-2 and TLR-4
Jossens et al. [121]	67 subjects with inactive and mild to moderately active CD	4 weeks	10 g oligofructose-enriched inulin (OF-IN) or placebo twice daily	Decrease in *Ruminococcus gnavus* and an increase in the number of *Bifidobacterium longum.* In the subgroup of patients with active CD, there was a positive correlation between the increase in the number of *Bifidobacterium longum* and the improvement in disease activity

## Data Availability

Not applicable.

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
