# Peer review of "The Role of Dietary Fibers in the Management of IBD Symptoms"

_nutrients, 2022, doi:10.3390/nu14224775_

Round 1

Reviewer 1 Report

1. line 315 assumption should be consumption
2. line 326 Patients should be small p
3. line 385 "because represents" should be because it represents
4. line 406 "Hallert et this group" maybe and his group
5. Table 1 typo in mitsuyama et al results box

Also the microbiome section I feel is a bit lacking. An important recent paper that should be mentioned is:
Targeted suppression of human IBD-associated gut microbiota commensals by phage consortia for treatment of intestinal inflammation (cell.com)

Author Response

Dear Reviewer 1, 

Thank you for your suggestions. Please find attached the answers to your comments. 

Reviewer 2 Report

This is a review on the role of dietary fiber in IBD management. There are some interesting and comprehensive observations in the review and in general it is a worthwhile manuscript. However, it is too long, often without a consistent focus on the topic.

1.       The introduction has the wrong focus. The first part of the introduction (epidemiology and etiology of IBD) is outside the scope of this review and may be shortened by 80%. On the other hand there is nothing in the introduction that gives the back ground for the main topic of the review. I suggest that the introduction can start with the paragraph starting with the words “Regarding environmental factors, there….” And focus this paragraph eventually on nutrition in general and then provide the rationale why you decided to focus on fiber and why this is particularly interesting (in reality make sections 5 and 6 the introduction, after shortening). Currently, the reader can start reading only from the “The aim of the present review” without missing even one important word for this review.

2.       Likewise, I suggest deleting sections 2.1, 2.2, 2.3, 2.4, 3.1, 3.2, 3.3, 3.4 and 4. This is not a review about IBD but on fibers on IBD. OK to provide 2-3 sentences of pathophysiology if necessary to understand something related to the fiber but currently this is a mixture of a general review on IBD with some on fiber. Start with section 5.

3.       In the retained section, the content is not well organized and there are many repetitions. Try to reorganize the content of each section so that it contains information only on that section without trying to cover more tangential topics.

4.       You are missing a discussion on the novel findings from Edmonton (several abstracts published) that some specific fibers are anti inflammatory and some not.  

5.       The English is poor in many sentences and the manuscript would enjoy the touch of a professional language editor.

6.       “After 12 months in all 414 three groups there was a failure remission maintenance, therefore the three therapies 415 showed the same efficacy. Thus, Plantago ovata seeds may be as effective as mesalamine 416 in maintaining remission in ulcerative colitis [94]. The”- the authors mix failure of superiority trial with non-inferiority design. The conclusion of the authors here is thus wrong. Especially given the very small sample size in each group and the fact that both interventions were not superior to either alone.

Author Response

Dear Reviewer 2, 

Thank you for your suggestions. Please find attached the answers to your comments. 

Reviewer 3 Report

Dear authors,

The manuscript submitted by Claudia Di Rosa and their colleagues evaluates the role of dietary fibers in CD and UC.

Some issues must be clarified before consideration for a possible publication. Please find my suggestions and comments below:

1.     A synthesis of the etiopathogenetic factors of IBD should be presented in a table.

2.     2.2. Pathogenesis: it would be better to add more references for this paragraph

3.     A table comparing the signs and symptoms of CD and CU should be added

4.     More evidence regarding pharmacological therapy of CD and UC should be provided

5.     Some evidence of side effects of DF should be provided.

Author Response

Dear Reviewer 3, 

Thank you for your suggestions. Please find attached the answers to your comments. 

Round 2

Reviewer 2 Report

Thanks for addressing my comments,